# Gerontology and Geriatrics in Undergraduate Nursing Education in Portugal and Spain: An Integrative and Comparative Curriculum Review

**DOI:** 10.3390/healthcare12171786

**Published:** 2024-09-06

**Authors:** Sara Brás Alves, Carlos Pires Magalhães, Adília Fernandes, Mª José Fermoso Palmero, Helder Fernandes

**Affiliations:** 1Instituto Politécnico de Bragança, Campus de Santa Apolónia, 5300-253 Bragança, Portugal; sarabras@ipb.pt; 2Research Centre for Active Living and Wellbeing (LiveWell), Instituto Politécnico de Bragança, Campus de Santa Apolónia, 5300-253 Bragança, Portugal; cmagalhaes@ipb.pt (C.P.M.); adilia@ipb.pt (A.F.); 3Universidad de Salamanca, Campus Viriato, Escuela de Enfermería de Zamora, Av. de Requejo, nº 33, 49022 Zamora, Spain; euemjfer@usal.es

**Keywords:** global aging, geriatric, gerontology, nursing education

## Abstract

Nurses play a critical role in caring for elderly patients; however, the emphasis on aging care in undergraduate programs may be insufficient. The present study aims at identifying the relevance given to theoretical and/or practical gerontological and geriatric contents in undergraduate study plans in Portugal and Spain. Presenting a two-part investigation, an integrative review approach examines nursing education on a global scale and a comparative analysis, using Bereday’s comparative method, to assess the nursing curricula between Portugal and Spain. The search found 117 documents, with 16 being included. Studies covered diverse educational practices in geriatric and gerontological nursing, emphasizing curriculum development, faculty expertise, practical training, attitudes towards elderly care, and future directions. The comparative analysis of nursing curricula revealed that Portugal places a priority on building foundational theoretical knowledge in the first year and then gradually integrating practical training. In contrast, Spain emphasizes an extensive and integrated approach with a strong focus on practical skills and comprehensive assessments. Our research emphasizes the need to incorporate aging-focused education into nursing curricula and update the curriculum, providing hands-on training with early exposure to these environments. Additionally, simulation classes can enhance critical thinking by allowing students to experience aging effects firsthand.

## 1. Introduction

The current global demographic landscape is characterized by an elevated average life expectancy, diminished birth rates, and a growing number of older adults [1]. The processes that define and impact aging are complex [2]. Biologically, the process of aging is characterized by the gradual accumulation of diverse forms of molecular and cellular damage [3,4]. Over time, this damage leads to a gradual decrease in physiological reserves, an increased risk of chronic diseases, and a general decline in the capacity of the individual [5]. However, these changes are not only inconsistent but are also not directly linked to age in years [3]. This occurs because numerous mechanisms of aging are arbitrary, but moreover due to the fact that these changes are significantly influenced by the environment and the behaviors of the individual [5].

In nursing, it is important to differentiate between geriatrics and gerontology when caring for the elderly. Geriatric nursing primarily focuses on providing care for elderly patients who are unwell or afflicted by identifiable conditions. Being medically oriented, it concentrates on diagnosing and treating health issues commonly affecting the population. It involves direct patient care and managing diseases frequently associated with aging, senility, and senescence [6,7]. On the other hand, gerontological nursing has a wider scope, emphasizing holistic care of older adults with a focus on promoting healthy living and overall well-being. This sphere integrates preventive measures, health promotion, and holistic care approaches to enhance the quality of life for older adults. It addresses not only the medical needs but also the social, psychological, and emotional aspects of aging [8].

Although, in the past, families were mainly responsible for looking after the elderly, nurses have always shown a keen interest in caring for the elderly, taking on this responsibility [8]. However, societal changes have transformed this model, making professional caregivers a crucial part of elderly care [9]. As the population of older adults grew, so did the recognition of the importance of specialized care for this demographic. During the 19th century, there was a notable advancement in the formalization of nursing as a profession. It was during this period that increased attention was devoted to the care of older adults. Florence Nightingale played a significant role in promoting comprehensive care for the elderly, emphasizing not only their physical health but also their emotional, social, and spiritual well-being [10]. During the 20th century, efforts were made to formalize nursing education and to professionalize the field significantly [11]. This era saw the introduction of specialized gerontologic training for nurses, aimed at recognizing and meeting the unique needs of the elderly [12]. As time passed, the approach to caring for older adults shifted towards a more holistic perspective, with leading organizations such as the World Health Organization calling for improved healthcare practices for aging populations [13]. Nurses, as frontline healthcare providers, are strategically positioned to address the complex health needs of elderly patients [14]. Conversely, the increasing demand for specialized care is not being met with a corresponding interest among nursing students [15]. The lack of emphasis on gerontological and geriatric units in nursing education programs leads to many students graduating without a thorough understanding of the challenges associated with aging [16]. The implications of this deficit are even more profound. By considering gerontological and geriatric nursing as a secondary area and privileging the study of surgical areas or specialties such as pediatrics, and obstetrics, the stigma that working with older people is a “less important” job and that nursing older people is poorly valued in society and among nurses perpetuates the devaluation of such an important work [16]. It is crucial to evaluate and rethink the incorporation of aging-related content in nursing studies to ensure that future nurses are adequately prepared for the challenges of an aging society.

It is important to note that the present work is part of a major project, entitled “Long-Lived Societies Project”. This is a collaboration between Portugal and Spain, being our mission to address longevity as a societal strength by promoting scientific research, technological advances, and behavioral practices across both nations. Understanding that, in Europe, the Directive 2005/36/EC of the European Parliament and of the Council of 7 September 2005 establishes and regulates the minimum training limits for general care nurses in the European Union, with theoretical and practical teaching necessarily focusing on various topics, including “*Care to be provided to the elderly and geriatrics*” [17], our research team questioned: What is the relevance given to theoretical and/or practical contents of gerontological and geriatric, in nursing degree study plans, between Portugal and Spain?

The central objective of our study is to analyze and compare the incorporation of geriatric and gerontological content within nursing curricula in Portugal and Spain—the two countries participating in the project mentioned. By synthesizing data from a range of curricular sources and educational literature, proposing to provide a comprehensive understanding of the current state of nursing education regarding aging care.

To better understand the nursing curricula in Portugal and Spain, we applied Bereday’s methodology. This approach was chosen primarily because Bereday, despite not being widely recognized, grounds his research and work development in a multidisciplinary context, drawing significant influence and consistently emphasizing that social, political, and geographical contexts play a crucial role in shaping educational practices [18]. Bereday’s comparative method systematically analyses the curricula’s content, structure, and practical training components [19,20]. This will contribute to identifying gaps and strengths in the curricula, guiding future curriculum development, and informing policy recommendations to enhance the preparation of nursing students for the care of elderly patients.

## 2. Materials and Methods

This study is conducted in two distinct but complementary segments: an integrative review and a comparative curricular analysis.

The first segment, the integrative review, systematically examines the existing literature on gerontology and geriatric nursing education globally, setting a foundation for understanding the broader context in which Portugal and Spain’s nursing programs operate.

This review was conducted following the five-stage method outlined by Whittemore and Knafl’s methodological model [21]. This approach allows for the inclusion of diverse articles, including empirical qualitative and quantitative studies and theoretical reports. By accommodating a variety of perspectives, this method is particularly valuable for advancing nursing science and practice. The framework consists of five steps: problem identification, literature search, data evaluation, data analysis, and presentation [21].

The second segment, the curricular analysis, involves a detailed comparison of the gerontology and/or geriatric nursing curricula from selected nursing schools in Portugal and Spain. Using the four steps of Bereday’s comparative method [19,20], we systematically focused on identifying disparities and similarities between the two countries. First, we begin by individually describing the data from the nursing curricula of both countries (referred to by Bereday as the description phase). After that, we interpret the data to understand the context and significance of the curricular elements (interpretation). We compare the described curricula in the juxtaposition phase to identify similarities and differences. Finally, a simultaneous analysis is conducted based on the comparison in the fourth step (corresponding to comparison) [19,20].

This study was not registered.

### 2.1. Problem Identification

The problem under investigation concerns the presence of aging-related content in nursing education programs. Specifically, this study seeks to explore how nursing programs prepare students to care for an aging population, the gaps that may exist in current curricula, and the educational strategies employed to enhance geriatric nursing competencies. By examining existing literature, this work seeks to provide a comprehensive understanding of the current state of aging education in nursing and identify areas for improvement.

### 2.2. Literature Search

A literature review is critical for appraising and analyzing a research field. According to Whittemore and Knafl’s method, the search phase should incorporate as many sources as possible, ensuring transparency and a thorough explanation of each sample decision as critical components of this phase [21]. The study was conducted by dividing the search into two specific segments to ensure comprehensive data collection and organization. The first segment collected peer-reviewed articles, empirical studies, and theoretical discussions on aging-related content in nursing education, published in Portuguese, Spanish, and English, with no specific timeline. The search was conducted via PubMed, CINAHL (via EBSCOhost), Nursing Allied (via EBSCOhost), and MedLine (via EBSCOhost). The leading search terms included “nursing education”, “geriatric nursing”, “nursing curriculum”, and “gerontology education” (Appendix A). The study selection process started by identifying and compiling all records uploaded into Rayyan^®^ (Qatar Computing Research Institute, Doha, Qatar), a research tool where duplicate citations were systematically removed. The remaining records were then screened based on their titles and abstracts to exclude those irrelevant by the primary reviewer (SA) and scrutinized by two reviewers (CM, HF), and any disagreements were discussed between the reviewers. Once studies were identified for possible inclusion, the full texts of selected studies were reviewed by two reviewers (SA, HF). Articles that appeared relevant or ambiguous were retrieved in full text for further evaluation. Any disagreements were discussed between reviewers, and a third reviewer (AF) was involved to help reach a consensus, as required. Grey literature was searched following the same process. To ensure rigor in the review process, the Preferred Reporting Items for Systematic Reviews and Meta-Analyses (PRISMA) guidelines were followed for this review’s identification, screening, eligibility, and inclusion stages [22,23].

The second segment aimed to gather official curriculum documents, course descriptions, and educational frameworks from nursing programs and institutions in Portugal and Spain. A representative sample of nursing schools from both countries was selected. We included institutions of varying sizes, geographical locations, and educational emphases. In Spain, we included schools actively involved in the Long-Lived Societies Project and those that frequently collaborate with our institution. In Portugal, we selected some of the largest public nursing schools, including institutions from the mainland and the islands, to ensure broad geographical representation. Additionally, we included schools that have established collaborative relationships with our institution, facilitating smoother data collection and richer insights.

Official curriculum documents, course descriptions, and syllabi from the selected institutions were obtained through institutional websites. This involved searching the websites of nursing schools and universities for publicly available curriculum documents and exploring guidelines from nursing accreditation bodies like the Ordem dos Enfermeiros (OE) and the Consejo General de Enfermería (CGE). In Portugal, we have selected the following institutions: (i) Instituto Politecnico de Bragança–Escola Superior de Saúde de Bragança (ESSa), (ii) Escola Superior de Enfermagem do Porto (ESEP), (iii) Escola Superior de Enfermagem de Coimbra (ESEnfC), (iv) Universidade da Madeira (UMa), and (v) Universidade dos Açores (UAc). In Spain, our chosen institutions include: (i) E.U. de Enfermería de Zamora, University of Salamanca (USAL), (ii) the Universidade de Santiago de Compostela (USC), (iii) the Universidad Complutense de Madrid (UCM), (iv) Universitat de lles Illes Baleares (UIB), and (v) the Universidad de La Laguna (ULL). Curriculum documents were analyzed to identify the inclusion of aging-related topics, and the depth and breadth of content related to senior care were evaluated. The findings between Portugal and Spain were compared to identify disparities in curriculum development, highlighting best practices and areas needing improvement.

### 2.3. Data Extraction

A standardized data extraction form was systematically employed to facilitate the comprehensive information collection from each study. This extraction tool, as meticulously detailed in Appendix A, underwent necessary adjustments and thorough review for every source of evidence encompassed in the review. In addition, the data extraction process meticulously examined curriculum documents and course syllabi to discern the depth of senior care content and critical findings from various institutions. The outcomes of this process are outlined in the results section.

### 2.4. Data Evaluation

To conduct a thorough quality assessment, it is essential to evaluate the relevance and rigor of the research [21]. To ensure a comprehensive evaluation, specific appraisal tools tailored to the respective research methods were used to assess the included studies in the literature review segment of this study. The Mixed Methods Appraisal Tool (MMAT) [24,25] was employed to appraise the quality of quantitative, qualitative, and mixed-methods studies. This critical appraisal tool is specifically designed for systematic mixed-study reviews, enabling assessing methodological quality across five categories of studies. The JBI Critical Appraisal Checklist for Systematic Reviews and Research Syntheses [26] was also used to evaluate systematic reviews (Appendix A). Furthermore, the JBI Critical Appraisal Checklist for Textual Evidence [27] was also applied to assess expert/option documents (Appendix A).

Two researchers (SA and CM) conducted the assessment to ensure a comprehensive and rigorous quality appraisal. Any discrepancies that arose during the appraisal process were thoroughly discussed between the reviewers, and in cases where consensus was not reached, a third reviewer (HF) was available to provide input and facilitate resolution.

It is important to note that studies were not excluded based on quality. Aligning with the recommendation by Whittemore and Knaf [21], it was considered that if studies contributed with valuable insights, these would be integrated.

### 2.5. Data Analysis

After collecting data from these different segments (Literature Review and Nursing Curricula), the information was integrated and synthesized during the data evaluation and analysis phases. This process involved organizing the collected information into relevant themes for the study and identifying patterns, commonalities, and differences across the various types of literature. The findings from the comparative analysis were integrated to provide a comprehensive understanding of how each curriculum prepares nursing students for geriatric care. Data were organized into tables to visually represent the comparisons and findings, including comparative tables showing the distribution of ECTS credits (European Credit Transfer and Accumulation System), practical training hours, and key curriculum components.

## 3. Results

### 3.1. Included Studies Characteristics

The initial search identified 117 citations. Following the removal of duplicates, 101 unique citations were identified. After the title and abstract review, 43 studies were assessed for inclusion, and 26 suffered full-text review. Studies were excluded if they did not delve into specific aspects of gerontological nursing education. Studies were excluded if they did not specifically focus on healthcare professionals in gerontological nursing education. Articles that did not address aspects of gerontological nursing education and those solely focused on general nursing education without emphasizing gerontology were also excluded. Furthermore, articles lacking clear educational or practice-related outcomes relevant to gerontological nursing were excluded to ensure actionable insights into the effectiveness of educational strategies and their impact on practice. The Preferred Reporting Items for Systematic Reviews and Meta-Analysis (PRISMA) diagram outlines the review process and search outcome [22] (Appendix A).

Our comprehensive review revealed a wealth of recent research from diverse global perspectives, with 16 studies included for analysis. The papers included in the research were published up to 2024, with no initial time restrictions applied to the search timeframe. They were sourced from various countries, including South Africa [28], the USA [29,30,31], South Korea [32], Iran [33], Taiwan [34,35], Belgium [36], Saudi Arabia [37], China [38], Portugal [39], Austria [40], Finland [41], and Uganda [42]. The papers covered a wide range of educational practices in geriatric nursing. The studies included qualitative and quantitative research, educational program evaluations, and surveys to explore and address challenges in geriatric nursing education. The main findings are in the Appendix A.

The following section presents the essential findings and implications of the studies reviewed on nursing education in geriatric care. The results are presented thematically to offer insights and strategies for improving educational practices to meet the growing demands of aging populations worldwide.

#### 3.1.1. Curriculum, Faculty Expertise and Practical Training

In the specialized field of geriatric nursing, an array of scholarly studies has brought to the header the pressing requirement for comprehensive, standardized curricula and the development of faculty expertise. Researchers, including Naidoo et al. [28], Hsieh and Chen [34,35] have emphasized the importance of uniformity in geriatric nursing curricula. Their findings and recommendations center on the critical necessity of conducting independent assessments, clearly defining and communicating standards, and ensuring consistent clinical placements to ensure comprehensive coverage of geriatric topics. Moreover, Krichbaum et al. [29] and Ghaffari et al. [33] have drawn attention to the shortage of faculty possessing profound insight into geriatrics. They have outlined the need for robust faculty development initiatives, emphasizing the importance of fostering faculty expertise to effectively implement curricula and adequately prepare nursing students for the complexities of geriatric care.

Furthermore, the significance of practical training and clinical experience has also been underscored. Ghaffari et al. [33] and AlSenany and AlSaif [37] have emphasized the deficiencies in practical training opportunities and the lack of suitable clinical practice settings for nursing students, particularly gerontology. They advocate for increased hands-on experience and the presence of specialist educators to enhance the teaching of geriatric care.

#### 3.1.2. Attitudes towards Older Care

Negative attitudes towards older adults have been emphasized, and there is a clear need for sufficient practical training and clinical exposure in the nursing field. Scholars like Chang and Do [32] and Xu et al. [38] have drawn attention to the existence of ageism among nursing students and faculty. Their research emphasizes the significance of implementing educational approaches to tackle these prejudices and foster empathy within the healthcare sector. Furthermore, Chang and Do demonstrate the effectiveness of innovative techniques such as storytelling, person-centered care, and technological advancements in shaping nursing professionals’ attitudes and fostering empathy towards older adults [32].

#### 3.1.3. Future Directions

The GEROM programme, as delineated by Brunner and Kada [40], exemplifies the significance of international collaboration in shaping culturally adaptable and comprehensive geriatric nursing education programmes. Such initiatives play a crucial role in fostering a global perspective and ensuring high-quality education in gerontological nursing. Furthermore, the research presented key recommendations for curriculum reform, policy changes, and future research directions.

Hsieh and Chen [34] and Naidoo et al. [28] (2020) underscored the importance of policy interventions to integrate geriatric nursing content into nursing curricula. They also emphasized the necessity for collaboration between educational institutions and policymakers to establish clear guidelines and standards.

Krichbaum et al. [29] underscored the significance of continuous faculty development programs to enhance nurse educators’ expertise in geriatrics, ensuring they remain abreast of the latest knowledge and skills.

Additionally, there was a call for longitudinal studies to assess the long-term impact of geriatric nursing education, advocated by Koskinen et al. [43], and Bevil et al. [31] also supported the integration of interdisciplinary core curricula across various healthcare professions to promote collaborative care models and holistic management of older adults.

### 3.2. Institutions and Curricula

To thoroughly compare geriatric nursing curricula at specific institutions in Portugal and Spain, we have followed the four steps of Bereday’s comparative analysis. This analysis aims to evaluate essential aspects and findings in geriatric nursing curricula comprehensively. The main findings are in the Appendix A.

The Gerontological and Geriatric Nursing course is offered in the first year and second semester at ESSa, and it is worth 3 ECTS. The main objective of this curriculum is to introduce students to the fundamental terminology and concepts associated with aging. The course aims to provide an understanding of the aging process from historical, cultural, and social perspectives. It delves into the biological and psycho-social changes that occur with aging, distinguishing between normal aging processes and pathological deviations. The course also addresses stereotypes and prejudices in gerontology, with the goal of fostering a more empathetic and accurate understanding of elderly care. Additionally, it emphasizes best practices in gerontological and geriatric care and identifies the policies and support networks available for the aging population. By the end of the course, students are expected to understand the complexities of aging and be capable of differentiating between normal and abnormal aging processes, thus laying a solid theoretical foundation for future practical applications in their nursing careers.

The course at USAL is offered in the third year and first semester and is worth 6 ECTS. It provides a comprehensive and practical understanding of geriatric nursing, starting with an introduction to the demographic and sociological profiles of individuals over 65 in Spain. The curriculum covers general aspects of geriatric nursing and gerontology, including the concepts of gerontology and geriatrics, different situations of old age, such as frailty, and the structural and physiological changes associated with aging. It also includes a thorough exploration of aging theories. A significant portion of the course is dedicated to a comprehensive approach to the geriatric patient, involving global geriatric assessments that cover physical, functional, cognitive, and social aspects. This practical focus extends to health promotion and preventive activities, such as vaccinations, exercise, and daily life care. Moreover, the curriculum addresses common geriatric pathologies and syndromes, such as immobility, pressure ulcers, falls, dementia, delirium, constipation, incontinence, and polypharmacy. It also covers the most critical diseases in the elderly and discusses the levels of care, social resources, and supportive products available to assist elderly patients.

While the curriculum at USAL does not explicitly detail its key findings and objectives, it is evident that the course aims to equip students with the practical skills and knowledge necessary for effective geriatric nursing. The focus on comprehensive geriatric assessments and the management of common pathologies suggests an objective to prepare students for the complexities of elderly care in a practical, hands-on manner.

Comparatively, the curriculum at ESSa focuses more on providing a solid theoretical foundation in the early stages of nursing education, covering fundamental concepts and fostering a comprehensive understanding of the aging process and the specific needs of older people. On the other hand, the curriculum at USAL is designed to offer a broader and deeper exploration of elderly care, combining theoretical knowledge with extensive practical skills.

The course unit “Adult and Elderly Health” at ESEP during the first year, first semester (6 ECTS), covers a historical perspective of adult health, including theories and concepts, biological, psychological, and social characteristics of adults, and the epidemiological transition. It also addresses health risk management, diseases preventable by vaccination in adults, the adult vaccination schedule, healthcare-associated infections, occupational risks, and the promotion and prevention of accidents and co-morbidities. In addition, it includes the assessment of worker health, aging concepts and theories, demography and epidemiology of aging in Portugal, aging as a developmental transition, factors influencing the health status of the elderly, geriatric syndromes, and integrated continuing care. The curriculum also explores the impact of terminal illness and death on individuals and families, euthanasia, dysthanasia, and orthothanasia, family care, adult and geriatric/gerontological assessment, and nursing interventions to promote adult health and active aging. The learning objectives include recognizing the characteristics of adult and elderly development, identifying the influence of professional factors on workers’ health, understanding labor legislation related to safety, hygiene, and health, identifying physiological changes related to aging, describing the vaccination schedule for adults and the elderly, identifying changes related to adult and elderly sexuality, understanding the cultural and social meaning of the dying process, recognizing Ministry of Health programs for adults and the elderly, planning health promotion and disease prevention actions, and developing autonomy and decision-making skills to address health problems in adults and the elderly.

In the third year, second semester at USC, the course unit “Life Cycle Nursing: Geriatrics” (4.5 ECTS) consists of two modules. The first module covers the basics of geriatrics and gerontology, including conceptual bases, aging of populations, demography of old age, gerontological evaluation, cognitive assessment, psycho-affective assessment, and social valuation. The second module focuses on geriatrics, including anatomical-functional changes in aging, their impacts on health, pharmacology in the elderly, and various geriatric syndromes such as cognitive impairment, dementia, acute confusional syndrome, infections, pressure ulcers, falls, post-fall syndrome, immobilization syndrome, and urinary incontinence. The learning objectives are: to understand the demographic, social, and health impacts of population aging; to know the levels of elderly care; to differentiate between aging characteristics and diseases in the elderly; to identify factors that influence or alter elderly health; to comprehensively understand the geriatric patient and assess their needs; to provide care to the elderly and their families while teaching disease prevention and health recovery behaviors; to plan nursing care considering the mechanisms, manifestations, and evolution of pathologies in old age; and to integrate into the healthcare team for planning, executing, and evaluating elderly care plans. Both institutions aim to prepare students for providing comprehensive geriatric care. However, they have different focuses based on their educational philosophies. ESEP emphasizes a broad understanding of adult and elderly health, including historical perspectives, risk management, preventative measures, and comprehensive end-of-life care, with the goal of developing skills in health promotion, disease prevention, and personalized care. On the other hand, USC provides a detailed exploration of geriatrics and gerontology, concentrating on the anatomical, functional, and cognitive aspects of aging. This program highlights the identification and management of geriatric syndromes and the integration of elderly care into the healthcare team.

The curriculum at ESEnfC covers topics such as social configurations, demographic trends, support networks, geriatric syndromes, principles of care for advanced illness, comprehensive geriatric assessment, and active aging strategies. In contrast, UCM’s curriculum extensively discusses aging demographics, theories, biological processes, major geriatric syndromes, ethical issues, pain management, and palliative care. Madrid’s curriculum is broader, with a strong emphasis on ethical considerations and practical care planning. At ESEnfC, the curriculum focuses on elderly nursing and geriatrics, which is offered in the second year and first semester and is worth 3 ECTS. The primary objective of this course is to recognize the challenges and opportunities associated with demographic, epidemiological, and social transitions at both international and national levels. The curriculum covers several vital areas, including aging and society, focusing on current social configurations and demographic trends, as well as support networks and social support for older people and families. Concepts and theories of aging are addressed, detailing the aging process and changes in primary aging. The curriculum also delves into geriatric syndromes, discussing frailty, postural instability, falls, malnutrition, sleep disorders, infections, sphincter incontinence, and cognitive and communication impairments. It highlights available technologies and services that enhance the safety and protection of older people and their careers. Principles of care for older people with advanced illnesses and at the end of life are included, along with comprehensive geriatric assessment and nursing interventions. The curriculum also addresses violence and maltreatment of older people. It promotes active aging through nursing intervention strategies that encourage healthy lifestyles, health surveillance, and the management of therapeutic regimes. The aim is to develop competencies in autonomy and decision making to solve health problems for older people, plan health promotion and disease prevention actions, and understand barriers that older people and their families face in accessing healthcare.

At UCM, the focus on geriatric care is integrated into two distinct course units. The first course, Community Nursing II, is offered in the third year, first semester, and is worth 6 ECTS. This course covers the care of the elderly, focusing on healthy aging and care for patients in fragile or dependent situations. The main objective is to acquire skills necessary for nursing professionals in primary health care, emphasizing health promotion, prevention, and care for prevalent health problems in the community, both individually and for population groups. The second course unit, Nursing of Old Age, Palliative Care, and Pain Management, is offered in the fourth year, first semester, and is also worth 6 ECTS. This course addresses the demographics of aging and its socio-health repercussions, the biological process and theories of aging, and the manifestations of aging in the elderly. It includes comprehensive geriatric assessments, the atypical presentation of diseases in the elderly, and the study of major geriatric syndromes such as dementia, frailty, gait disorders, falls, dysphagia, incontinence, depression, and isolation. The curriculum also covers the most prevalent health problems in the elderly, ethical issues in elderly care, the pathophysiology of pain, types of pain, and pain treatment. It emphasizes palliative care, addressing the various symptoms of terminally ill patients, and strategies for accompanying bereavement. Both institutions intend to prepare students for comprehensive geriatric care by combining theoretical knowledge and practical application. However, ESEnfC focuses heavily on understanding societal changes, theoretical concepts of aging, and specific geriatric syndromes to establish foundational knowledge and decision-making skills early in the nursing program. In contrast, UCM offers a more detailed and practical approach spread across two curricular units, emphasizing community nursing, health promotion, management of common health problems, as well as palliative care and pain management in the elderly. ESEnfC’s curriculum is designed to foster a broad understanding of geriatric care early on, while UCM’s curriculum provides a more specialized and practical focus in the later years of the nursing program. Both approaches are aimed at providing students with the requisite skills to effectively address the complex needs of the aging population.

The “Adult and Elderly Nursing” course at UAc, offered in the second year, first semester, and worth 6 ECTS, focuses on promoting healthy aging, understanding community resources, and developing relational skills and personalized care. It covers aging and longevity, explores biological, psycho-social, and cultural factors, and emphasizes strategies for promoting healthy aging and longevity with quality. The course includes topics such as the rights of adults and the elderly, prevention of stereotypes, work and retirement issues, accident prevention, prevention of ill-treatment, urinary and intestinal elimination problems, physical exercise, prevention of complications from immobility, sexuality, sleep and rest, communication with individuals with sensory impairments and dementia, and spirituality and religiosity. Additionally, it highlights the importance of community resources and social policies and their implications for nursing practice. The primary objectives are to describe aging and longevity, identify strategies for promoting healthy aging and quality longevity, and recognize the implications of existing policies and community resources. On the other hand, ULL provides a comprehensive approach with its “Clinical Nursing I—Gerontogeriatric Nursing” course, which takes place in the second year, first semester, and is worth 6 ECTS. The course is divided into four main parts, focusing on clinical nursing assistance for adults with health issues, nursing care for older adults, workshops and seminars, as well as simulation exercises. The first part deals with chronic illnesses, care in hospital settings, and the assistance of hospitalized and surgical patients. The second part specifically focuses on gerontology and geriatric nursing, including topics such as aging and elderly health issues, in-depth geriatric assessment, and caring for elderly patients experiencing loss, grief, and death. The third part involves practical workshops and seminars on fundamental clinical techniques, nursing assessment of the elderly, strategies for promoting healthy aging, and care for dependent individuals. The fourth part offers simulation exercises on essential nursing procedures such as medication administration, bladder catheterization, nasogastric catheterization, wound care, oxygen administration, secretion aspiration, and peripheral venous catheter placement. The curriculum aims to develop a thorough understanding of clinical and gerontogeriatric nursing, to plan and provide nursing care using standardized nursing languages and taxonomies, to perform nursing techniques safely and effectively, and to foster a reflective and critical approach to societal influences on clinical and gerontogeriatric nursing. Moreover, students are encouraged to independently explore new content within the field. Both institutions emphasize the importance of promoting healthy aging and providing quality care for the elderly. UAc focuses on theoretical knowledge and practical strategies for healthy aging, emphasizing the role of community resources and relational skills. Meanwhile, ULL offers a more hands-on approach with practical workshops and simulations, fostering critical thinking and technical skills in gerontogeriatric nursing. Both curricula aim to equip students with the knowledge and skills necessary to effectively address the complex needs of the aging population.

At UMa, the curriculum includes three specific units focusing on geriatric nursing, two of which involve clinical practices. The course “Nursing II—Nursing for the Elderly”, offered in the first year, second semester, and worth 3 ECTS, covers the aging process, theoretical perspectives, bio-psycho-social-cultural and spiritual implications, as well as stereotypes and prejudices associated with aging. It also addresses social and health policies, community resources, promotion of healthy aging, autonomy and empowerment, and nurse intervention aimed at enhancing the quality of life. The curriculum emphasizes the humanization of care, support for caregivers, prevention of complications, dignification of the end of life, and elderly-centered integrated care. The main objectives are to provide knowledge about the aging process, adapt nursing care to the needs of older adults, support elderly people in developing their health projects, and acquire relational and technical skills for personalized and quality care. The second unit, “Elderly Care Practice I”, offered in the second year, second semester, and worth 12 ECTS, focuses on the practical application of nursing care for the elderly. Students plan, execute, and evaluate nursing interventions using the nursing process as a scientific methodology and the taxonomy of the International Classification for Nursing Practice (ICNP). The aim is to care for the elderly as individuals in transition, striving for well-being and independence in both health and disease contexts. The third unit, “Adult and Elderly Care Practice II”, also in the second year, second semester, and worth 18 ECTS, extends the practical experience to comprehensive nursing care for adult and elderly patients with medical and surgical conditions in a hospital context. This unit emphasizes the integration of technical, scientific, and relational dimensions in nursing interventions, focusing on developing technical and relational skills essential for nursing practice in medical and surgical environments. In contrast, the UIL offers a more theoretical and extensive curriculum in its course unit “Nursing the elderly”, provided in the second year, second semester, and worth 6 ECTS. The curriculum is structured into thematic units covering general gerontogeriatric considerations, social and health care, comprehensive assessment of the elderly, and general care for the daily life of the elderly. Topics include the concepts and consequences of aging, health care settings, abuse of older people, promotion of autonomy, home care, nutritional needs, physical exercise, sexuality, sleep physiology, pharmacotherapy, immune system changes, elimination issues, pain management, and cognitive problems such as dementia and Alzheimer’s disease. The course emphasizes the application of theoretical and methodological principles in professional practice, health education projects, and the use of scientific, technological, and technical knowledge to ensure the continuity and complementarity of nursing care. Specific objectives include applying theoretical principles in practice, assessing and planning health situations, and using scientific knowledge to support nursing care, with a strong focus on problem-solving and decision-making skills.

#### 3.2.1. Interpretation

The described courses offer significant insights into the differing approaches and emphases within each country’s educational framework.

For example, the Gerontological and Geriatric Nursing course at ESSa introduces fundamental terminology and concepts related to aging early in the nursing education process. It aims to build a strong theoretical foundation by covering the aging process from various perspectives, including biological, psycho-social, and cultural aspects. This curriculum emphasizes understanding normal versus pathological aging and addresses age-related stereotypes, which helps to foster a more empathetic and accurate perspective on elderly care. This approach reflects an early focus on theoretical knowledge, which prepares students for future practical applications.

In contrast, the course at USAL offers a more advanced and practical approach, focusing on comprehensive geriatric assessments and management of common geriatric conditions. By including a detailed exploration of aging theories and practical skills, this curriculum aims to provide students with hands-on experience and a thorough understanding of the complexities of geriatric care. This reflects a later-stage, more applied approach in the nursing education timeline. Similarly, the “Adult and Elderly Health” course at ESEP covers a wide range of topics related to adult and elderly health, including historical perspectives, risk management, and comprehensive care strategies. This curriculum integrates theoretical concepts with practical applications, aiming to develop broad competencies in health promotion and disease prevention.

The “Life Cycle Nursing: Geriatrics” course at USC emphasizes a detailed understanding of geriatric and gerontological principles, with a focus on the assessment and management of elderly health issues. The curriculum is designed to integrate theoretical knowledge with practical skills, preparing students to address complex geriatric needs effectively. Comparatively, the curriculum at ESEnfC covers theoretical concepts related to aging and geriatric syndromes, with an emphasis on societal and policy aspects, while the UCM curriculum provides a comprehensive and practical approach to geriatric nursing, focusing on community care and palliative care.

At UAc, the course on “Adult and Elderly Nursing” integrates theoretical and practical aspects of healthy aging, community resources, and personalized care. Meanwhile, ULL’s “Clinical Nursing I—Gerontogeriatric Nursing” offers a hands-on approach with practical workshops and simulations, emphasizing technical skills and critical thinking.

Lastly, the UMa curriculum combines theoretical knowledge with extensive practical experience in nursing care for the elderly, focusing on a holistic approach to aging and quality of life. In contrast, UIL’s “Nursing the Elderly” offers a more theoretical focus, with an emphasis on applying knowledge to professional practice and decision-making skills.

#### 3.2.2. Juxtaposition

Table 1 below aims to outline the commonalities and distinctions between the curricula of Portugal and Spain.

#### 3.2.3. Comparison

In Portugal, the allocation of ECTS credits for geriatric nursing courses typically varies from 3 to 6 credits. For example, ESSa offers 3 ECTS, focusing on foundational theoretical knowledge in the first year, while other institutions like ESEP and ESEnfC provide 6 ECTS, covering a broader range of topics and integrating practical training with theoretical content. The Portuguese approach prioritizes establishing a strong theoretical foundation early in the program and gradually introducing practical elements.

In contrast, Spanish geriatric nursing courses often allocate 6 ECTS, reflecting a more extensive and integrated approach. Programs at institutions such as USAL, UCM, and ULL emphasize a detailed exploration of geriatric care, with a strong focus on practical skills and comprehensive assessments. Spanish curricula typically include a substantial amount of credits for practical applications, highlighting the importance of hands-on experience in preparing students for real-world scenarios. This comprehensive approach extends through detailed coverage of geriatric syndromes, palliative care, and preventive health measures.

In terms of content focus, Portuguese curricula emphasize fundamental theoretical concepts of aging early in nursing education, aiming to provide students with a solid understanding of aging from historical, cultural, and social perspectives. In contrast, Spanish curricula offer a more detailed and practical approach to geriatric care, integrating extensive content on aging theories, comprehensive assessments, and practical strategies for managing common geriatric pathologies.

Regarding clinical practice, Spanish programs place a strong emphasis on hands-on experience through workshops, simulations, and clinical placements, allowing students to apply theoretical knowledge in real-world settings. Portuguese programs also include clinical practice, but the extent and integration of this practice vary across institutions. The focus in Portugal tends to be on developing a theoretical understanding initially, with practical training introduced progressively throughout the program.

## 4. Discussion

During our literature review, a significant difference in terminology was noticed. While “geriatric nursing” is listed as a MeSH (Medical Subject Headings) term, “gerontological nursing” is not. This highlights the important distinction between geriatrics and gerontology, and how they each impact the care of the elderly. Such substantial disparities and a noticeable lack of recognition in this field have been noticed before, in the 1970s. As such, geriatric nurses sought help from the American Nurses Association (ANA) to promote the profession. This collaboration led to comprehensive studies, recognizing the importance of nursing in specialized elderly care and its role in promoting healthy aging and maintaining well-being. As a result, the profession’s title was reconsidered, and “gerontological nursing” was chosen as the most suitable term to encompass nursing care in this area [8,44]. However, through these terms, we were able to notice significant differences in the approach to aging education in nursing, both in our literature review and in the curricular analysis of nursing schools in Portugal and Spain.

Specifically, European studies tend to focus on gerontological nursing, while studies from other regions primarily concentrate on geriatric nursing (see Appendix A). The distinction between gerontological and geriatric nursing is evident not only in the terminology used but also in the focus areas of research and the resulting discoveries. In particular, the contributions of Naidoo et al. [28], as well as Krichbau et al. [29], Hsieh and Chen [34], and Xu et al. [38] are all related to geriatric nursing and emphasize the need for improved educational programs to develop the necessary clinical and technical skills needed to address the diseased patient and forthcoming healthcare challenges in geriatric and long-term care settings. Consequently, nursing education in these regions may be more inclined towards preparing students for the clinical aspects of elder care, emphasizing the diagnosis, treatment, and management of chronic and age-related health problems. On the other hand, European studies, such as those by Deschodt et al. [36] and Tavares et al. [39], underline the need to enhance gerontological nursing to positively impact older adult care through health promotion and a holistic approach. Such divergence in focus might be attributed to varying healthcare priorities and educational philosophies.

In Europe, there appears to be a stronger weight on patient-centered and preventative care, which aligns with the principles of gerontological nursing [45]. The distinctions in nursing education across different regions serve as a critical reminder of the need to comprehend the underlying philosophies that shape these variations thoroughly while considering that the healthcare needs of the elderly are often shaped by the priorities of healthcare systems [46,47].

It is important to note that although Portugal and Spain are close geographically, they have unique cultural backgrounds and different lifestyles, possibly leading to differences in their perspectives on nursing care for older adults. These variations are seen in their core nursing education—where Portugal emphasizes gerontological nursing, and Spain prioritizes geriatric nursing—and in the structure of their programs and specialized opportunities after getting a license. Such particularity is of significant interest, specifically when considering that one of the objectives of the Bologna Process, instituted in 1999, was to unify nursing curricula across Europe [48]. Naidoo et al. [28], Hsieh and Chen [34,35], also established the necessity for collaboration between educational institutions and policymakers to establish clear guidelines and standards for nursing curricula. This aligns with the recommendations of ANA, which emphasizes that nurses, or future nurses working with the aging population, require specific competencies to provide the highest quality of care. Among these competencies is the ability to holistically assess older adults, considering their physical, emotional, mental, social, spiritual, and functional status [45]. Despite efforts, there remains a need for standardization and enhancement of gerontology content in nursing education [39,49]. Developing a standard gerontology curriculum and competencies can positively impact older adult care and influence the next generation of nurses [50].

The distinct approaches in the nursing curricula of Portugal and Spain provide valuable insights into the preparation of nursing students for gerontological and geriatric care. The Portuguese emphasis on theoretical knowledge and comprehensive understanding of aging equips students with a broad perspective on elderly care, while the Spanish focus more on practical skills and hands-on experience, ensuring that graduates are ready to manage complex geriatric cases effectively. Both approaches have their merits and together highlight the importance of a balanced nursing education that integrates both theoretical and practical components. These go hand in hand with the studies found, which highlight the need for a balanced approach that combines strong theoretical knowledge with extensive practical training to prepare nursing students effectively [33,37].

However, not all institutions uniformly address this importance. Several Portuguese and Spanish schools lack specific internships dedicated to clinical practice with older patients. Practical experience is essential for integrating theoretical knowledge with clinical skills, achieving successful outcomes, and patient safety [51,52]. As such, providing adequate resources, proper guidance, and sufficient time for practice are essential when educating future nurses [51,53,54].

It is important to note that clinical practice will not only have an influence on the retention and application of knowledge but also on the promotion of evidence-based (good) practices. Authors Chang and Do (2024) [32] and Wenxian Xu et al. (2024) [38] highlight the prevalence of ageism in nursing education and that students often hold negative attitudes towards older adults, associating aging with physical and mental decline [32,38]. Studies also identified that nursing students reported that gerontological and geriatric education was often misrepresentative, as older persons were only mentioned in the context of disease and disability and not normal aging [55,56,57].

These misconceptions can have significant implications for the quality of care provided to elderly patients, leading to suboptimal care practices, reduced patient satisfaction, and poorer health outcomes for older adults [58,59].

One effective approach to challenging these preconceptions is to incorporate earlier internships, for example in nursing homes, as a form of clinical practice. Research has demonstrated that increased interactions with older adults lead to more positive attitudes toward aging and the elderly [60]. As such, by engaging with elderly patients in a real-world setting, students can gain a more nuanced understanding of aging, recognizing that it is a natural process that varies significantly among individuals [37] who have unique needs and capabilities rather than being characterized solely by decline [3].

Simulation classes, where students experience the effects of aging firsthand, can also be a powerful tool in nursing education [61]. These experiences often involve using age-simulation suits to replicate the sensory and physical limitations that come with aging [61,62]. Participating in these experiences can greatly enhance students’ empathy by giving them a better understanding of the challenges faced by older adults, such as reduced mobility, impaired vision, and hearing loss [55]. However, while these simulations can enhance empathy, they need to be carefully managed to avoid reinforcing ageist stereotypes.

Research has identified a dual outcome in this context: students who concentrate on the limitations may unintentionally reinforce negative perceptions [63,64]. Therefore, it is essential to complement simulation exercises with comprehensive debriefing sessions and discussions that highlight the strengths, resilience, and diverse experiences of older adults [63].

Noteworthy, educational programs in both Portugal and Spain incorporate strategies to address negative attitudes and promote empathy, including modules on aging concepts and theories, demography, and epidemiology of aging, emphasizing the importance of understanding the cultural and social meanings of the dying process.

Healthcare services increasingly prioritize the aging population, emphasizing the crucial need for nurses to develop expertise in this area [65]. Recognizing the prevalence of complex health conditions among the elderly underscores the crucial requirement for nursing professionals to possess a comprehensive range of skills to assist with the diverse needs of this demographic. Nurses should demonstrate a variety of core competencies, including clinical expertise, critical thinking, effective communication, cultural proficiency, leadership skills, ethical decision making, and a dedication to continuous learning. 

Sustained professional development is essential to ensuring that adult nurses stay current with the latest healthcare developments and sustain their proficiency throughout their careers. This constant learning and skill enhancement are fundamental for delivering effective care to elderly patients [66]. Accordingly, the evidence advocates for the integration of gerontogeriatric nursing as a pivotal component in all nursing curricula [67]. It is worth noting that this field has been recognized for over 20 years, with the WHO Regional Office for Europe (2003) developing a gerontological nursing curriculum focused on preventive care and lifelong learning for nurses specializing in gerontology [68]. Both curricula prioritize health promotion and disease prevention, aligning with the global recognition of preventive care’s importance in enhancing the quality of life for the elderly. This shared emphasis reflects a broader trend in nursing education, aiming to prepare students to promote overall health and well-being in aging populations.

Examining the gerontological and geriatric nursing curricula in Portugal and Spain reveals notable differences in educational emphasis, which are reflected in the allocation of ECTS and course content. It is important to recall that ECTS credits reflect the workload required to complete course units, which includes lectures, practical work, seminars, private study, and examinations. Typically, one ECTS credit corresponds to 25–30 h of work. While examining the nursing education landscapes of Portugal and Spain through the lens of their respective ECTS allocations and curriculum focuses, a significant disparity emerges that reflects varying approaches to gerontological and geriatric nursing.

In Portuguese institutions, there is a lack of consensus regarding the number of ECTS assigned to theoretical courses, which range from 3 to 6 ECTS. For example, ESEP allocates 6 ECTS, while IPB and ESEnfC only allocate 3 ECTS. This variability suggests an inconsistent approach to the depth and breadth of theoretical knowledge provided to students. Conversely, Spanish nursing schools demonstrate a greater emphasis on geriatric education and a more substantial commitment to these subjects, with the lowest ECTS allocation being 4.5 ECTS. Such disparity in the allocation of European Credit Transfer and Accumulation System (ECTS) credits, as well as the integration of practical training, has far-reaching implications for the education of future nurses. The varying emphasis on these subjects across different institutions can significantly impact the competency and preparedness of nursing graduates to effectively care for an aging population [15,69]. In schools with lower ECTS allocations and limited practical training opportunities, students may not receive adequate exposure to the complexities of elderly care [70].

As indicated in the literature, the insufficient time dedicated to gerontological nursing in the curriculum can hinder their ability to proficiently manage gerontological and geriatric health challenges in their future careers [71]. This circumstance needs to be revisited and altered, as demands for older adults’ care are drastically increasing [72,73]. It is expected that more and more of these professionals will care for older adults, and there is a growing organizational awareness of the necessity for nurses to have expertise in this area [74].

## 5. Conclusions

The increasing global aging population presents significant challenges for healthcare systems, highlighting the urgent necessity for specialized geriatric and gerontological nursing. With the growing number of older adults, healthcare systems, along with educational institutions, specifically those training future nurses, must evolve to meet their diverse and complex needs.

Our findings emphasize the critical importance of incorporating aging-related content into nursing curricula to prepare future nurses better, encouraging nursing schools to revise their programs. It is essential to update existing curricular units to cover gerontological and geriatric care comprehensively, provide practical training opportunities focused on elderly patient care, and promote evidence-based practices in gerontology and geriatric nursing.

Placing a high priority on clinical placements and internships in settings like nursing homes, geriatric wards, and community care facilities is vital. These environments showcase a significant level of clinical complexity and offer substantial learning opportunities for students. Early exposure to these settings can help students gain hands-on experience and develop the necessary skills to effectively care for older adults. This could include simulation classes where students experience the effects of aging firsthand and debriefing sessions to discuss these experiences, enhancing individual critical thinking, empathy, and understanding of aging as a diverse and positive aspect of life rather than merely a period of decline.

Furthermore, these data can empower stakeholders to push for uniform curriculum standards, guaranteeing that every nursing graduate is fully equipped to address the needs of an increasingly elderly population.

## Figures and Tables

**Table 1 healthcare-12-01786-t001:** Similarities and differences between Portuguese and Spanish’ curricula.

	Portugal	Spain
ECTS	3 ECTS: ESSa, Uma 6 ECTS: ESEP, UAc, ESEnfC	4.5 ECTS: ESC 6 ECTS: USAL, UCM, ULL, UIL
Content Focus	ESSa: Fundamental concepts of aging, historical, cultural, and social perspectives ESEP: Adult and elderly health, aging concepts, epidemiology, health risk management ESEnfC: Social configurations, demographic trends, geriatric syndromes Uma: Aging process, bio-psycho-social implications, healthy aging, quality of life	USAL: Comprehensive geriatric care, demographic profiles, common pathologies, health promotion UCM: Community nursing, healthy aging, palliative care, comprehensive assessments <br> ULL: Clinical nursing assistance, workshops, practical skills in gerontogeriatrics ESC: Basics of geriatrics, anatomical-functional changes, geriatric syndromes
Clinical Practice	ESEP: Includes practical training and application of health risk management ESEnfC: Focuses on practical care principles and geriatric syndrome management UAc: Emphasizes practical application of healthy aging and community resources Uma: Practical application of aging concepts and interventions	USAL: Comprehensive approach including health promotion and preventive activities ULL: Extensive practical workshops and simulation exercises ESC: Practical workshops and seminars on fundamental techniques UCM: Extensive practical care planning and community-based activities

## Data Availability

The original contributions presented in the study are included in the article/Appendix A, further inquiries can be directed to the corresponding author/s.

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
