# Peer review of "Gerontology and Geriatrics in Undergraduate Nursing Education in Portugal and Spain: An Integrative and Comparative Curriculum Review"

_healthcare, 2024, doi:10.3390/healthcare12171786_

Round 1

Reviewer 1 Report

Comments and Suggestions for Authors

Dear authors,

The subject of the article is very important and relevant in light of the demographic changes and the increase in life expectancy. I have no doubt that the gerontological aspects are very important in geriatric care.

As a gerontologist who has been teaching courses on aging in many departments for the past decade, including in the undergraduate and graduate nursing department, I am a witness to the importance of the knowledge and information taught in these courses. This contributes, among other things, to raising awareness of this population as well as psychosocial and family aspects that are critical for optimal geriatric care.

I'll start by saying that the article was difficult to read for two reasons, both are very important. First, the text in each part is written in most parts as one mass and not divided into paragraphs. Dividing the text into short paragraphs would have helped reading and understanding. Second, the article presents two main topics which are related to each other: 1. Review of articles 2. Review of courses and study programs.

As far as I understand, the main goal of the study was to examine the relevance given to study units on aging in nursing education in Portugal and Spain, with the aim of identifying any gaps in the development of curricula in this field.

In order to simplify the article so that it would be easier to read and to focus on the main goal, the reviewed articles could have been presented as part of the introduction and focused on examining the study programs. Unfortunately, the review of the programs is also very cumbersome to read and therefore loses its potential contribution.

The article should contribute to the examination of study programs in nursing and their development, so it is very important to present the findings clearly.

Specific points for correction and improvement:

1. The title of the article reflects only the literature review and does not refer to the examination of the programs in Portugal and Spain.

2. Uneven and sometimes confusing use between gerontogeriatric, geriatric and gerontology. The subject must be refined in order to convey the message of the importance of gerontological aspects in geriatric care.

3. In line 192 it is stated that the reviewed articles are from 1988 - there is room for thinking whether it is indeed still relevant, since the field has much developed in the last two decades.

4. In lines 193-195, the countries where the studies used in the review were conducted are indicated. There is a very large variation between the countries and there is room for thinking about whether it is correct to base it on for the main purpose of the study.

5. In lines 211, 216, 217, delete the year of publication of the article and add its number in the bibliographic list where it is missing.

6. Part 3.1.3 is missing or there is a mistake in the numbering

I hope that this review will contribute to the improvement of the article so that it can achieve its goal and contribute to the field of nursing.

Reviewer 2 Report

Comments and Suggestions for Authors

Reviewer 3 Report

Comments and Suggestions for Authors

First of all, I must thank you for the opportunity to review this work.

It is an interesting review, focusing on a specific aspect of the education of future nursing professionals.

In the introduction, the authors could take the opportunity to expand on the concepts of geriatric and gerontological nursing, as it seems a bit brief. Perhaps you could include the historical and temporal evolution of the paradigm of care for the elderly. It would be beneficial to expand the bibliographic references on this subject.

We are informed that "this integrative review aims to examine the relevance given to curricular units on aging in nursing education in Portugal and Spain, seeking to identify any disparities in curriculum development in this area." However, the review includes not only the curricula of different nursing schools in both countries, but also articles on geriatric and gerontological nursing education from around the world.

The search strategy should be included in the article and not presented as supplementary material.

It is not clear how the universities whose curricula were analyzed were selected or why these were considered the most representative.

In the results section, there is an extensive comparison of the curricula analyzed, comparing characteristics in pairs, but the analysis of results from other documents is not as detailed and could provide more information (although it is true that the supplementary material is abundant).

I consider it a good piece of work that would benefit from small modifications.

---

Reviewer 4 Report

Comments and Suggestions for Authors

1.      Please justify why the study compared Portugal and Spain

2.      Please describe how representative sample of nursing schools was done

3.      What model has been employed to compare geriatric nursing curricula in Portugal and Spain.

4.      Please provide ethical considerations for obtaining curriculum documents, course descriptions

5.      Please try to use Aging population and older adults instead of elderly population and elderly individuals respectively

6.      Please tabulate the results

7.      Please describe integration process of the data (how integration has been conducted)

8.      It is suggested to use Bereday F G method to examine curricular units

9.      Please provide practical implication of the study

10.   Please provide full form of abbreviations such as ESSa and ECTS

Round 2

Reviewer 1 Report

Comments and Suggestions for Authors

I am glad that my comments contributed to the improvement of the article.

One aspect that was not addressed in the revised paper is the division of the text into paragraphs. As I mentioned earlier, writing the text as one essay without dividing it into short paragraphs makes it difficult to read and understand.

Author Response

Thank you for your valuable feedback. We appreciate your thorough review and comments.

We apologize for overlooking the aspect of dividing the text into shorter paragraphs.  Thank you for bringing this to our attention. We have revised the manuscript to ensure that the text is appropriately divided into shorter, more manageable paragraphs.